# Developing a Sustainable Omnichannel Strategic Framework toward Circular Revolution: An Integrated Approach

**Tuğba Yeğin** [1] and **Muhammad Ikram** [2,*]

1 Faculty of Economics and Administrative Sciences, Demir Celik Campus, Karabuk University, 78050 Karabuk, Turkey
2 School of Business Administration, Al Akhawayn University in Ifrane, Avenue Hassan II, P.O. Box 104, Ifrane 53000, Morocco
* Correspondence: i.muhammad@aui.ma

**Abstract:** One of the contributions of digitalization to cyclical change is the adoption of Omnichannel Marketing (OM) as a new marketing strategy for brands. In this research, we examined whether the quality of integration (INQ) in omnichannel environments has an effect on brand equity (BE) and its dimensions (brand loyalty (BL), brand association and brand awareness (BAS), and perceived quality (PQ)) within the framework of a structural model. We aim to expand the limited number of INQ research areas. In this context, in the first stage of our research, we conducted an online survey consisting of three parts with the consumers of the Nike luxury sportswear brand, which is in 11th place in the global brand value ranking, residing in Turkey from the developing countries. In the second stage of the analysis, we performed CFA for scale reliability and validity. Crobach's alpha, AVE and CR values for all factors of the scale exceeded the threshold values in the literature. In addition, the goodness-of-fit values of the scale, which were checked for compliance with the research, exceeded the threshold values. In the third stage of the analysis, we performed SEM analysis to test the model of the study and the assumptions of the study. The SEM results of our research confirmed the assumptions established between INQ and BE and its components in the context of OM. SEM results revealed that INQ had the highest effect (0.93) on BAS and PQ and the least effect (0.86) on BL, and INQ affected BE with 0.90. The results of this research, which examines the predictors of brand equity and its components, offer implications for OM, INQ, BE subject areas that have not been empirically analyzed despite increasing knowledge and still having limitations in theoretical information. Our research is unique, as it is the first study to empirically examine the relationship between INQ and BE and its components in the context of OM. The research on omnichannel applications is quite limited. This study brings a conceptual extension to the literature on omnichannel strategies, INQ and OM, whereas they presented the necessary reasons for managers to provide INQ in an omnichannel environment in order to increase brand equity, with an empirical application. In addition, the most important benefit of this research is that it shows brand owners and managers and brand marketers a way to set up the omnichannel system toward circular revolution.

**Keywords:** sustainable omnichannel; strategic framework; circular economy; consumer behavior; sustainable development; integrated channel

## 1. Introduction

Brand equity, competitive advantage, global recognition, and high net worth are important because they bring global recognition as a competitive advantage and increase financial value. The number of studies investigating the antecedents of brand equity as an abstract indicator is a marketing concept that needs to be reexamined according to each new marketing strategy (Oh et al., 2020; Aaker, 1991). According to [1], while Apple is the highest technology brand in 2020, Amazon (USD 200.667 billion) is in second place, followed by Microsoft (USD 166.001 billion) with Nike (USD 42.538 billion) in 11th

place. Existing research on multichannel marketing suggests that brand equity can be strengthened by providing cross-channel service quality [2–4]. Although there are many multichannel studies investigating the effects of channel integration quality on brand equity (e.g., [5–15]), since the omnichannel field is very new, there are limitations, literature gaps on this subject and within the provision of INQ in omnichannel marketing. It is extremely important to investigate the changes that may occur in the brand value. Brand equity, as an intangible asset created by marketing activities, is the sum of values that increase or decrease the value of products and services offered by the company to consumers, depending on the brand's distinguishing characteristics such as name or symbol [16–18]. Brand loyalty (BL) is the repeated purchase of products from the same brand continuously and consistently in the future [19]. Perceived quality (PQ) is the perception formed in the minds of consumers regarding the quality of the brand's products [20]. The Brand Association (BA) states that it is the unique characteristics (in the minds of consumers) that distinguish a brand from other brands [21]. Brand awareness (BAW) is the consumer's capability to realize, separate, identify and call to mind the product category to which the brand belongs [16,22].

The pressure of the sustainable development plan applied to prevent global warming at a global level, protect the environment, purify the air from $CO_2$ emissions to zero waste on brands, businesses and marketers force them to change their marketing and management strategies in harmony with technology [23–32]. This change is expected to enable the production and marketing of products that are produced, distributed, priced, and promoted with less cost, less waste, and more recyclable raw materials and green energy [23–30,33]. In this context, it is important for cyclical change that brands provide maximum service quality with minimum environmental pollution, energy loss, and time-saving in the channels where they deliver their products to their customers [23,25,29,30,33]. In this context, the solution is now omnichannel strategies and the quality of inter-channel integration that is expected to be provided in omnichannel channels.

Omnichannel marketing (OM), which provides brands with sustainable growth opportunities across many channels, is increasingly preferred by more brands, including channel harmony beyond multichannel marketing [34–37]. During 2021, 0.14% of all multi-channel marketing sales were single-channel marketing, and 83% of them were omnichannel marketing orders [38]. These comparative figures show that it is significant for marketers, brand owners, managers, retailers and to understand and apply OM well [2].

Omnichannel marketing, which offers the advantage of offering brands products to their customers through many channels, differs from multichannel marketing in that all channels are evaluated as a whole, allows customers to like a product through the brand's online channels and receive it from the physical store, or to discover the product in the physical store and buy it online [39]. However, due to the reasons arising from the lack of coordination between these channels, customers can give up and return the products they purchased. At the same time, these returns cause an increase in carbon emissions from waste products at the local and global level [39,40]. As a way to avoid the potential harms of OM, which is increasingly being adopted by consumers, the researchers recommend that brands implement marketing strategies to maximize the level of inter-channel coordination [41–43]. There is a lack of research in the literature on whether brands' INQ dimensions (Channel–Service Configuration (CSC), Content Consistency (CC), Process Consistency (PC), Assurance Quality (AQ)) provision affects the brands' intangible assets (BE, BL, BAW, BA, PQ) in omnichannel environments [39,43]. However, further investigation of these intangible assets, which the brand is trying to acquire, protect and increase with tangible and moral sacrifices in the long term, for omnichannel environments will provide both theoretical and practical contributions. In this context, despite the increasing use of OM, we aim to narrow this gap in the literature with the empirical results of this study in the context of developing countries.

As a result, the studies investigating the impact of INQ on BE and its dimensions for omnichannel are insufficient in the literature, and most of them are conceptual [43–45].

Brand equity, as an intangible asset created by marketing activities, is the sum of values that increase or decrease the value of products and services offered by the company to consumers, depending on the brand's distinguishing characteristics such as name or symbol [16,17]. BL is the repeated purchase of products from the same brand continuously and consistently in the future [19]; PQ is the perception formed in the minds of consumers regarding the quality of the brand's products [20]; BA states that it is the unique characteristics (in the minds of consumers) that distinguish a brand from other brands [17]. BAW is the consumer's capability to realize, separate, identify and call to mind the product category to which the brand belongs [16,22].

There are some cases where companies already have the infrastructure in place. Customers are encouraged to register products online by many fast-moving goods brands. Providing consumers with the ability to return products instead of scrapping them, as Apple already does, could increase the culture of recycling. Meanwhile, consumer power—driven by social media—could influence manufacturers to take circular actions [25,29]. Omnichannel service has emerged since consumers turn to brands that offer this service and desire to store across many channels simultaneously. OM studies are therefore based on consumers (consumer behavior, customer loyalty, purchasing behavior, etc.). For the last 12 years, researchers have been carried out in this examining the relations [43,46–49].

Channel synchronization enables products to be delivered across multiple channels of the same quality to consumers [36,50,51]. That is, it is essential to integrate all channels for OM to achieve its purpose [43,49,52]. INQ and factors have been discussed in previous studies as a predictor of Omnichannel strategies, customer, purchase, brand, service quality, perceived value and fluency, brand equity, customer loyalty (Table 1).

**Table 1.** Summary of omnichannel strategies related to past studies.

| Research | Omnichannel | Customer | Purchase | Brand | Service Quality | Perceived Value | Perceived Fluency | INQ | Effect on | Conceptualizing | Moderator | Analysis |
|---|---|---|---|---|---|---|---|---|---|---|---|---|
| [53,54] | ✓ | | ✓ | | | | | ✓ | Online Perceived Value and Online Purchase Intention | | | SEM |
| [54] | ✓ | | | | | | | ✓ | Perceived Service Quality and Perceived Risk | | | Diffusion Theory and Scenario Analysis |
| [55] | ✓ | | | | | | | ✓ | | Determinants of Cross-Channel Integration of Retailers | | Regression |
| [56] | ✓ | | | | | | | ✓ | Customer Satisfaction | | | |
| [43] | ✓ | ✓ | ✓ | | ✓ | ✓ | | ✓ | Cross buying intentions and perceived value | | | |

**Table 1.** *Cont.*

| Research | Omnichannel | Customer | Purchase | Brand | Service Quality | Perceived Value | Perceived Fluency | INQ | Effect on | Conceptualizing | Moderator | Analysis |
|---|---|---|---|---|---|---|---|---|---|---|---|---|
| [57] | ✓ | ✓ | ✓ | ✓ | | | | ✓ | Customer engagement | | | PLS-SEM |
| [58] | ✓ | ✓ | | | ✓ | ✓ | ✓ | ✓ | perceived fluency | | | SEM |
| [59] | ✓ | ✓ | | | ✓ | | | ✓ | consumers' satisfaction and loyalty | | | SEM |
| [45] | ✓ | ✓ | ✓ | | | | | | Customer Equity | | | Literature Review |
| [42] | ✓ | ✓ | | | ✓ | | | ✓ | | Service Quality Measurement for Omnichannel Retail: Scale Development and Validation | | SEM |
| [60] | ✓ | ✓ | | | | | | | Customer Loyalty and Mediating Role of Customer Engagement and Relationship Program Receptiveness | | | PLS SEM |
| [41] | ✓ | ✓ | | | ✓ | | | ✓ | Customer Loyalty | | Omni-Channel Shopping Experience | SEM |
| [61] | ✓ | ✓ | | | ✓ | | | | Customers' Attitudinal Loyalty and Positive Affect Experience | | | PLS SEM |

✓indicates the indicator has been used in prior studies.

However, there are aspects of this research that differ from the articles given in Table 1. First, this research is quantitative and tests assumptions with Structural Equation Model (SEM) analysis. Second, this study claims that INQ is a predictor of BE and its components. This claim was previously implicitly claimed in the 2017 qualitative study by Hossain et al., but this is the first time it has been examined with direct and quantitative analysis. As a result, the studies investigating the impact of integration quality (INQ) on brand equity (BE) and its dimensions for omnichannel are insufficient in the literature, and most of them are conceptual [43–45].

Although studies on the definition, dimensions, impacts, and strategies of OM affect the retail and service sectors in many ways, there is limited evidence in the literature on the impact of the quality of channel integration, which should be carefully considered in an omnichannel system, on brands and consumers [43,57,62]. Studies on the effects of the

high level of inter-channel integration quality of the brands that perform OM that will bring value to the brand are quite insufficient. Although the compatibility of multiple channels of a brand with each other forms the basis of OM, most of the studies on OM do not focus on INQ and do not observe how the components of brand equity are affected by INQ. Statistical reports are revealing that there are some brands that have lost share value and lost sales turnover as a result of failure to integrate [12,13,43]. These financial impacts on INQ's brand are immediate, but it may take longer to see the change in consumer-based brand equity. Brand equity is more intangible than financial values, in other words, the loyalty of the customers to the brand, the degree of recognition of the brand, the perceived quality of the brand, etc. With the realization of the total change in the components and the sub-components associated with these components, a change in brand value can be seen. However, with a relational model to be established between INQ and brand equity and its components, and testing this model with empirical analysis, the effects that may occur on the value of the brand in the absence of INQ can be presented to marketers and brand managers to take precautions. Moreover, the research of [45] suggested that future studies investigating the relationship between INQ and brand equity, brand image, knowledge, and loyalty by creating a scale and a data-oriented model are recommended, which is the reason for the emergence of this research. Moreover, this research relies on the research of [43], which conceptualizes the dimensions and sub-dimensions of INQ and empirically tests the data using dynamic capability theory to incorporate brand equity dimensions [43,50,51]. In this context, in this research, we establish a model between INQ and brand equity and its components, and within the framework of this model, we make four assumptions and reduce this gap in the literature by performing an empirical analysis.

In this research, we develop a model that provides experimental proof for the conceptional dimensions, presents the relationship between INQ and BE and its sub-dimensions (BL, PQ, BA /BAW) for omnichannel, and seeks answers to the following questions:

a.  What is the relationship between INQ and BE for OM?
b.  What is the relationship between INQ and the dimensions BL, PQ, and BA /BAW for OM?

In this context, in this study, we aim to discuss the effects of INQ on BE, BL, BAS, PQ and the results from this effect. For this, we develop a scale consisting of three parts, which is suitable for the purpose of our study, from the literature. The first part of this scale consists of the items that we aim to obtain for the demographic information of the participants. The second part consists of the scale items of INQ, and the third part consists of the scale items of BE and its components. Sources of [43] for the INQ scale (21 items) and [22] for brand equity (10 items) were used. We did not develop scales but simply adapted the scales from these two widely cited studies to our research. However, we tested the accuracy and reliability of these scales in the methodological part of the research. We used a 5-point Likert scale to measure the items. We had help from the statistics company, which has the second largest consumer portfolio in the country, to apply this survey to Nike sportswear consumers residing in Turkey. The company applied our online survey to 1549 customers, which we selected randomly from 198,000 Nike brand customers within its body. To our survey, 847 consumers responded. Of these, survey data were excluded from participants under 18 years of age, those who had no experience with all Nike brand channels, and those who provided inconsistent responses to the three attention control questions (ACQs). As a result, survey data from 626 participants were valid for the final analysis. We did not detect any response bias in the data of 626 participants. Thus, the number of participants in our study exceeded the lower limit of the literature that it is appropriate to carry out with five times as many samples as the number of scale items for SEM. Thus, we first tested the reliability and validity of the scale of our research with the data obtained. We applied Cronbach's alpha analysis, which is widely used as a basis for the measurement of scale reliability. Cronbach's alpha values of all factors were above the desired value of (0.70) in the literature. Second, we performed (confirmatory factor analysis) CFA to see the validity of our scale and its compatibility with our research. All

factor loadings of INQ ranged from 0.66 to 0.97, and for multiple correlation squares ($R^2$), values ranged from 0.44 to 0.94, and t > 2.58 was significant at the *p* < 0.001 significance level. All factor loadings of BE were significant between 0.79–0.88, and multiple correlation squares ($R^2$) values were between 0.62–0.77, and t > 2.58, *p* < 0.001 significance level. These values showed a strong relationship with the corresponding structures, in compliance with the literature [63–65]. In addition, AVE and CR values were important for us in terms of the reliability of the constructs. AVE values of 0.80 and CR values of 0.50 were obtained for all structures for the above-predicted values (Appendix A, Table A1) [64,66]. Thus, the CFA results and Cronbach's Alpha results show that the scales are consistent with research and are accurate and reliable. Then, the Pearson correlation analysis results showed that we examined the significance of the relationships between the first-order sub-dimensions of INQ (independent variable) and the first-order sub-dimensions of BE (dependent variable). Thus, the results showed that the correlation coefficients between the relation of INQ and its sub-dimensions and BE and its dimensions were statistically significant (*p* < 0.01) [67]. In addition, we obtained the discriminant validity of the scale structures with the results of discriminant analysis. The discriminant validity of our scale, which provided the necessary literature values for [68], was verified. The SEM results of our research confirmed the assumptions established between INQ and BE and their components in the context of OM. SEM results revealed that INQ had the highest effect (0.93) on BAS and PQ and the least effect (0.86) on BL, and INQ affected BE with 0.90.

The results of this research, which examine the predictors of brand equity and its components, offer implications for OM, INQ, BE subject areas that have not been empirically analyzed despite increasing knowledge and still having limitations in theoretical information. Our research is unique, as it is the first study to empirically examine the relationship between INQ and BE and its components in the context of OM. In addition, it is unique because it is the first study to examine the relationship between omnichannel strategies and brand equity in detail in the context of Turkey, a developing country with very limited studies on the Circular Revolution. Moreover, our research expands the limitations of the studies of [43,45].

In addition, with the results of this study, we present numerical data that provide cross-channel integration in omnichannel marketing that can be effective for marketers and brand managers who are looking for ways to increase the main brand value from the past. In other words, cross-channel integration can be used as a predictor of brand equity and its components. Brand managers who want to increase their brand equity can see the necessity of including INQ increasing strategies in their omnichannel strategies with the SEM results of this research. Thus, with this study, we not only shed light on future research in the field of OM, INQ, and BE, but also provide important advice to brand managers who carry out OM.

In this context, our research continues as follows: Section 2: Literature Review, Section 3: Exploratory conceptual framework and hypothesis, Section 4: Research methodology, Section 5: Results, Section 6: Discussion and implications, Section 7: Conclusions.

## 2. Literature Review

### 2.1. Omnichannel Marketing (OM)

In the literature, it is called "omnichannel marketing" when a consumer makes simultaneous purchases through many alternative channels such as physical stores, computers, smart devices (tablets, phones, wearable technology, kiosks, etc.), virtual stores with three-dimensional glasses, social media sites, and e-commerce platforms (Amazon, Alibaba.com, etc.), which includes searching for the product before purchase, researching product features, comparing multiple brands that make the same product, viewing the product in various physical and virtual retail locations, touching the product, purchasing and receiving the product, and returning it. All these channels and this great multichannel service are offered to consumers by marketers [69–73].

International research shows that OM, which is still evolving and reflects consumers' desires to store on multiple channels simultaneously, has been particularly adopted by

retailers [37,74–76]. OR provides retailers with an integrated retail experience by combining all the services [76,77].

OR not only enables the simultaneous use of channels but also integrates all existing channels through various digital technologies for a retailer or brand to provide great customer knowledge [43,77–79]. Customer touchpoints, i.e., all channels that are common areas for OM and OR, should be viewed and organized like the harmony of all musicians in an orchestra [80]. The need to expand the concept of INQ for omnichannel, multiply its examples, and adapt it to retailers in all industries is supported by many studies [43,60,81,82].

### 2.2. Integration Quality (INQ)

Previous research has found that INQ is a brand evaluation tool for consumers that influences them and has valuable and positive outcomes for marketers [60,83–85]. The concept of channel integration refers to the coordination between all customer touchpoints (websites, physical stores, kiosks, smartphones, etc.) of a brand or retailer to provide consumers with a sustainable and flawless experience [51,57,60,86]. INQ is considered in the literature as the performance of channel integration [43,45,57,58,60,87–91].

INQ is the ability to provide seamless service to customers across all channels, and its importance for omnichannel marketing has been repeatedly highlighted in the literature [51]. Brands can profit from a superior competitive advantage in the marketplace by integrating all the channels they use [92].

### 2.3. Dimensions of INQ

Research on INQ and its dimensions claim that a retailer's channels can have not only physical but also virtual quality, and cross-channel inconsistency can reduce overall quality perceptions [43,50,51,57]. The research in [51], among the phenomenon researchers who make this claim, argues that INQ is an important factor in omnichannel services so that consumers do not have problems buying services through more than one channel. Relevant research has confirmed the importance of INQ [43]. In this study, we used the research of [43] for the INQ dimensions (CSC, CC, PC, AQ). The detail description of previous studies that used different approaches and methods reflects the Omnichannel is presented in Table 2. The definitions of the dimensions are given in Table 3 in the next sections.

### 2.4. The Outcome of Integration Quality, Brand Equity, and Dimensions

INQ has been involved in some customer-focused studies in recent research in the Omnichannel and multichannel context.

**Table 2.** Related research methods, approach, analysis, implications and differences of this research belonging to the related research.

| Research | Factor | Approach | Analysis | Aim and Results of Research |
|---|---|---|---|---|
| [90,93] | - | qualitative | - | This study, which examines past research to examine the continuity strategies of OM qualitatively, examines mostly integration quality dimensions, horizontal and vertical strategies. |
| [94] | Promotion, Product and Price, Information Access, Order Fulfillment Customer Service, Transparency, Freedom, and Synchronization | quantitative | SEM Analysis | The study explores the customer experience in the OM environment. This study, which investigates the effect of Transparency, Freedom, and Synchronization dimensions on customer loyalty, shows that customer experience mismatch has a negative impact on customer retention, but channel transparency, ease and smoothness can effectively mitigate this negative impact. |

**Table 2.** *Cont.*

| Research | Factor | Approach | Analysis | Aim and Results of Research |
|---|---|---|---|---|
| [95] | Content Consistency, Marketing Consistency, Synchronization | quantitative | SEM Analysis | Expanding the knowledge in the field of customer experience and channel integration, this research revealed that the 4Ps are more effective than emotional in improving the cognitive customer experience. |
| [43] | Breadth, Transparency, Content Consistency, Process Consistency | quantitative | SEM Analysis | Focusing on the integration quality area in the context of OM, the study presents cross-buying intent as a result of INQ. |
| [96] | - | quantitative | MANOVA | This research, which revealed that the brand image can be increased due to the digital customer data obtained in the omnichannel environment, was built on shopping scenarios. |
| [97] | - | qualitative | - | The case study was conducted with 15 retail store managers. The results of the analysis revealed that the use of technology in physical stores strengthens brand loyalty and brand image. |
| [48] | Content Consistency, Process Consistency, Marketing Consistency, Freedom, Synchronization, Working Together | quantitative | SEM Analysis | The study carried out in the retail industry conceptualized the customer experience in the omnichannel environment as a multidimensional construct and found that perceived compliance and risk play a mediating role in the plot of omnichannel shopping intention with the omnichannel experience. |
| [45] | - | qualitative | - | In this study, which conceptualizes the integration quality dimensions for Omnichannel marketing and presents its impact on customer equality through a literature review, and INQ is seen as a predictor of customer equality. |
| This Research | Channel-Service Configuration (CSC), Content Consistency (CC), Process Consistency (PC), Assurance Quality (AQ) | quantitative | SEM Analysis | For OM, we explore the impact of INQ on brand equity and its dimensions with an empirical analysis for the Nike brand. We conceptualize INQ as a predictor of BE and its components BL, BAW, BA, PQ. |

Previous research has shown a relationship between service integration and customer equity [45,98]. However, there is no quantitative research in the omnichannel integration quality literature that uses data to examine the relationship between INQ and the drivers of customer equity. Customer equity is examined in the qualitative research of [45]. Here, the concept of equality is the feeling of getting what one deserves, which ensures mutual equity in relationships [45]. It is the feeling on which long-term relationships, especially between the brand and the customer, are based [99–101]. Customer equity, however, is the sum of the lifetime value of all customers of a company [86,102,103] and has three driving forces: value equity, relationship equity, and brand equity [104]. Value equity encompasses the customer's actual valuation of the brand and is their perception based on what they gave up when they purchased the product during their buying behavior [104]. Relationship equity deals with what the customer gets from the relationship based on the perceived total inputs in the relationship between a customer and a brand [105]. Brand equity is the sum of the customer's tangible and intangible perceptions of the brand. The brand's marketing strategies influence it. It has been used as part of customer equity since the 2000s. While brand equity can make the brand attractive to new customers, it contains the miraculous power to create repurchase behavior among existing customers and build an emotional attachment to the brand.

The formation of brand equity is related to the following four dimensions [16]: BL, PQ, BA, and BAW. Although brand equity has been studied since the 1980s, it has been a driving force in customer equity studies since 2000 [102,106]. Several multichannel studies have examined the relationship between multichannel service quality and brand equity [107–110]. However, in an omnichannel marketing environment, a brand may be more influenced by brand equity as part of customer equity than in a multichannel environment [45,52,71]. Since omnichannel allows customers to store on all channels simultaneously, there will be differences compared to brands that do not use omnichannel that will affect not only the customer's emotions but also the buying experience, relationship quality, perceived value, and brand equity [111]. In this context, [45] recommended further research on omnichannel marketing efforts and data-based analysis of customer equity drivers (value equity, relationship equity, brand equity). At the same time, [43] suggested that it would be beneficial to research the dimensions and impacts of INQ in different domains, consumer groups, and industries. Due to the lack of research focusing on these recommendations and their relationship with integration quality (INQ) and brand equity, there is an opportunity to measure the dimensions of brand equity (BL, PQ, BA, and BAW) as INQ outcomes.

## 3. Exploratory Phase: Conceptual Framework and Hypotheses Development

In the first phase, before quantitative data analysis with Confirmatory Factor Analysis (CFA) and Structural Equation Modeling (SEM), the INQ is composed of four main dimensions: Channel Service Configuration, Content Consistency, Process Consistency, and Assurance Quality. We determined them through qualitative data analysis based on articles [43,45,50,51,57,58,93]. In the second phase, before CFA and SEM, brand equity consisted of four main dimensions: BL, PQ, BA, and BAW. The dimensions of BA and BAW were combined under a single factor from these dimensions. Finally, brand equity was influenced by the qualitative data analysis of three main dimensions.

In the third phase, we discussed the accuracy of this qualitative data analysis results in online meetings with two professors, three associate professors, and researchers in omnichannel and brand equity in the institutes of three different countries with high q rankings.

In the last phase, we have established hypotheses that we will analyze with SEM using the extensive research we have conducted to determine the impact of INQ on brand equity and its dimensions, as well as the content of articles with a high number of citations scanned in Scopus and the Web of Science. Being aware that qualitative data analysis is the research's dynamics, we paid special attention to this phase. In this way, the research contributed to expanding information about INQ and omnichannel research areas and reproducing limited knowledge through qualitative data analysis.

This section aims to discuss the findings of the qualitative phase and the model's hypotheses presented in Figure 1, which were developed for the research of quantitative data analysis, with the supporting literature sources in the following sections.

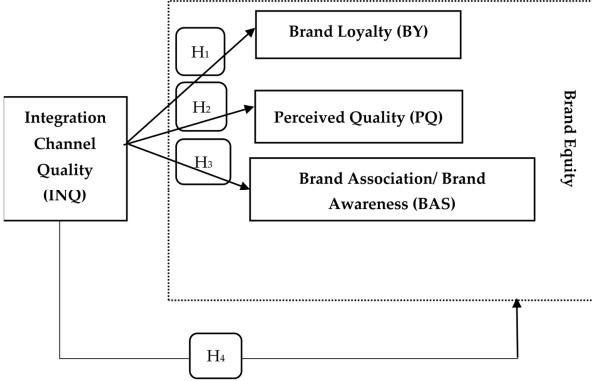

**Figure 1.** Sustainable omnichannel strategic framework.

### 3.1. Integration Quality (INQ) and Brand Loyalty (BL)

Consumer-created perceptions of service quality offered by the brand are believed to have a significant impact on actions such as brand satisfaction, repurchase, the recommendation to the surrounding community, and brand dependence [112–115]. Several multichannel studies have examined the relationship between multichannel service quality and BL [62,107,110,116,117]. Although there are many studies on INQ for OM, few studies have examined its effects on building BL [62]. BL exists when a customer repeatedly and consistently buys products from the same brand in the future [19]. With OM, which involves an integrated service approach across all channels, it may be more feasible than multichannel marketing to strengthen existing customers' relationships with the brand, thereby creating dependency among customers and thus ensuring that loyal customers repeatedly purchase the brand's products, i.e., creating BL [62,118–120]. In this context, we make the following assumption:

**H₁:** *Integration quality (INQ) positively influences brand loyalty (BL).*

### 3.2. Integration Quality (INQ) and Perceived Quality (PQ)

PQ is a dynamic of brand equity. Zeithaml (1988) defines PQ as the consumer's sense of a product's superiority. PQ can be influenced by factors such as the general sense of the consumer, their experiences with the product, their prejudices, their needs, their environment, the brand's offerings, their relationship with the brand, etc., and it is a competitive advantage for brands as well as more sales [121–127]

Despite a large body of literature on PQ [20,128,129], there are few studies on integration quality or the relationship between service quality and PQ [126]. The quality of services offered through brands' online and offline channels and customer satisfaction when shopping across all channels influence PQ [22,130–132]. The consumer realizes the brand's superiority when he feels high quality through prolonged product use [133]. Therefore, the consumer will prefer the brand's product that he perceives as having higher quality than other brands [20]. In this context, we make the following assumption:

**H₂:** *Integration quality (INQ) positively influences perceived quality (PQ).*

### 3.3. Integration Quality (INQ) and Brand Association/Awareness (BAS)

BA, defined as everything that appears in consumers' minds about a brand and whose importance for brand equity has been noted by brand authors [134] and [116], is combined with BAW [22]. BAW is the definition, meaning, and information about the brand in the consumer's mind [21]. BAW is the antecedent component of brand equity, and increasing consumer BAW leads to brand preference and differentiation from competitors [17]. Consumers' positive experiences with the brand result in positive BAW/BA. Furthermore, being able to shop from all brand channels can impact BAW/BA as a positive reflection of the same quality service and integration quality [135,136]. Several studies on multichannel marketing have proven that service quality impacts the BA and BAW [110]. Online and offline service quality perceptions combine with consumers' overall service perceptions to create a positive judgment for BAW/BA [137]. Consumers' positive experiences with shopping options and services offered by traditional and e-commerce sites lead to increased BAW/BA [133]. In this context, we make the following assumption:

**H₃:** *Integration Quality (INQ) positively influences Brand Association/Awareness (BAS).*

### 3.4. Integration Quality (INQ) and Brand Equity (BE)

Brand equity is the assets and identities associated with a brand, brand name, or symbols added to or subtracted from the value of products or services of a company or its customers [138]. While the research of [139] and [17] evaluate the concept of brand equity as the added value that the brand name adds to products, Ref. [140] defines it as consumer confidence in brand identity and image. Central to the brand equity literature definitions

is that brand equity combines BL, PQ, BAW, and associations [22,138]. In a sense, brand equity is an important structure for the brand that represents customers' evaluation of the brand, attracts customers to the brand, enables them to recognize and associate with the brand, and is influenced by its marketing strategies and 4Ps [17]. Customers' recognition of a brand can have a different effect on BAW by creating a brand's positive image in their memory. Positive, reciprocal, and long-term relationships between customers and the brand can be built due to high-quality and integrated service quality across all online and offline channels. If this relationship can be built, it can lead the customer to respond to the brand's marketing mix's 4P elements (product, price, place, promotion). The brand's feeling and positive perception of the product desired from all customer touchpoints, such as price, ease of payment, speed of delivery, campaign, special discount, customer service, and the purchasing process at the same level can enable BAW, superiority and direct and positive effects on brand equity [20,22,136,137]. This is because studies on omnichannel marketing, consumer attitudes, multichannel marketing, and service quality point in this direction [9,110,135,141–143]. In this context, we make the following assumption:

**H₄:** *Integration Quality (INQ) positively influences brand equity (BE).*

## 4. Research Methodology

### 4.1. Measures

As a result of the literature review, the research assumed that INQ had four first-order sub-dimensions: CSC, CC, PC, and AQ [43]. Furthermore, as a result of INQ, it was assumed that Brand Equity (BE) consisted of three first-order sub-dimensions: BL, PQ, and BAS (BAS abbreviation used instead of BAW/BA abbreviation) [22].

Sources of [43] for the INQ scale (21 items) and [22] for Brand Equity (10 items) were used. We did not develop scales but simply adapted the scales from these two widely cited studies to our research. However, we tested the accuracy and reliability of these scales in the methodological part of the research. We used a 5-point Likert scale to measure the items. Table 3 provides brief conceptual definitions of the structures that make up the scales, their scales, and the resources used in detail.

**Table 3.** Functionalization of structures.

| Structures | Definitions | Sub-Dimension | Definitions |
|---|---|---|---|
| INQ- [43] | It is the performance of a brand or retailer in coordinating all customer touchpoints (websites, physical stores, kiosks, smartphones, etc.) owned by the retailer to provide consumers with a sustainable and flawless experience [43,51,57]. | (CSC) | Channel performance in terms of the same quality and consistency across all channels [43,50,57,58]. |
| | | (CC) | It refers to the consistency of information going and coming through the brand's different channels [51,57,60]. |
| | | (PC) | Consistency of consumer elements (sense of service, wait time, image, level of employee appreciation) across channels [43,50]. |
| | | (AQ) | It refers to the security provided to consumers across all channels. ASQ encompasses the totality of privacy, security, and e-service quality and is conceptualized as a dimension of INQ [43,60]. |

**Table 3.** *Cont.*

| Structures | Definitions | Sub-Dimension | Definitions |
|---|---|---|---|
| BE- [22] | Brand equity is the assets and identities associated with a brand, brand name, or symbols added to or subtracted from the value of products or services of a company or its customers [138]. | (BL) | It is the repeated purchase of products of the same brand by a consumer in the future [16,21]. |
| | | (PQ) | It is a consumer's sense of a product's superiority [20]. |
| | | (BA) | Everything about a brand appears in the consumer's memory [21]. |
| | | (BAW) | It is the set of definitions, meanings, and information about the brand in consumers' memory [21]. |

*4.2. Data Collection*

Survey data were collected from March–January 2022 with the support of research companies with a database of approximately 198.000 consumers in Turkey. We had help from the statistics company, which has the second largest consumer portfolio of the country, to apply this survey to Nike sportswear consumers residing in Turkey. The company applied our online survey to 1549 customers, which we selected randomly from 4.800 Nike brand customers within its body. To our survey, 847 consumers responded. Of these, survey data were excluded from participants under 18 years of age, those who had no experience with all Nike brand channels, and those who provided inconsistent responses to the three attention control questions (ACQs). As a result, survey data from 626 participants were valid for the final analysis. We did not detect any response bias in the data of 626 participants. Thus, the number of participants in our study exceeded the lower limit of the literature that it is appropriate to carry out with five times as many samples as the number of scale items for SEM [63,144].

**5. Results**

*5.1. Demographic Results*

The demographic profile of the participants is presented as a percentage in Table 4.

**Table 4.** Demographic profile of the participants.

| Gender | | Marital Status | |
|---|---|---|---|
| Female | %50.2 | Single | %47.4 |
| Male | %49.8 | Married | %52.6 |
| **Education** | | **Age** | |
| High School and Below | %39.1 | 18–24 | %26.2 |
| University | %33.5 | 25–35 | %21.7 |
| Graduate | %27.3 | 36–44 | %26.4 |
| | | Over 45 years old | %25.7 |
| **Job** | | **Distribution Channels Used** | |
| Student | %12.6 | I buy the products of this brand online (from the website), they are delivered to my address. | %31.6 |
| Private Sector Employee | %37.2 | I buy products of this brand online (website), delivered from store. | %33.1 |
| Officer | %54.2 | I buy the products of this brand from the store; they are delivered to my address. | %35.3 |
| Academical Personal | %4.6 | | |
| Teacher | %7.2 | | |
| Small Business | %8.1 | | |
| Engineer | %3.2 | | |
| Military Officer | %2.2 | | |
| Nurse | %5.9 | | |
| Financial Advisor | %4.6 | | |
| Doctor | %5.0 | | |
| Lawyer | %5.1 | | |

According to Table 4, the ratio of male and female participants was almost equal. Likewise, the rates of single and married participants were very close to each other. The

majority of the participants were university and master's graduates. It was found that 26.2% of participants were 18–24 years old, 21.7% were 25–35 years old, 26.4% were 36–44, and 25.7% were 45 or older, while the majority of participants were classified as private-sector employees at 37.2% (n = 233). The channel usage alternative rates of the participants were very close to each other.

*5.2. Result for Measurement*

In this research, the reliability analysis and confirmatory factor analysis methods were first used to test the consistency and accuracy of the multilevel scales (INQ scale and brand equity scale) created using the literature. CFA specifies multifactorial structures and tests the fit of the model to the data [145]. We iteratively tested both scales to determine the goodness-of-fit values and to obtain a final model. In assessing goodness of fit, we followed the recommendations of [65] ($X^2 < 3$; GFI > 0.90; AGFI $\geq$ 0.900; NFI > 0.90; RMSEA < 0.01, 0.05 and values below 0.08 indicate excellent, good, and moderate fit, respectively; 0 < CFI < 1) which are widely accepted in the literature [65]. We also performed the Cronbach's alpha test for scale reliability.

We analyzed the validity and reliability of INQ constructs (CSC, CC, PC, AQ) and BE (BL, PQ, BAS) in two separate path diagrams and reported the significant values of the constructs in the same table (Table 5). All factor loadings of INQ ranged from 0.66 to 0.97, and for multiple correlation squares ($R^2$), values ranged from 0.44 to 0.94, and t > 2.58 was significant at the $p < 0.001$ significance level. All factor loadings of BE were significant between 0.79–0.88, and for multiple correlation squares ($R^2$), values were from 0.62–0.77, and t > 2.58, $p < 0.001$ significance level. These values showed a strong relationship with the corresponding structures, in compliance with the literature [63–65].

**Table 5.** Inter-order relations values.

|  | **X²/df** | ***p*** | **RMSEA** | **CFI** | **GFI** | **AGFI** | **NNFI** | **NFI** | **RMR** | **SRMR** |
|---|---|---|---|---|---|---|---|---|---|---|
| INQ | 1.932 | 0.000 | 0.039 | 0.99 | 0.97 | 0.97 | 0.99 | 0.99 | 0.017 | 0.013 |
| BE | 1.374 | 0.000 | 0.020 | 0.99 | 0.99 | 0.99 | 0.99 | 0.99 | 0.009 | 0.008 |

AVE and CR values were important for us in terms of the reliability of the constructs. AVE values of 0.80 and CR values of 0.50 were obtained for all structures for the above-predicted values (Table 6) [64,66].

**Table 6.** AVE and CR Values.

| **Variables** | **CR** | **AVE** | **Cronbach's Alpha** |
|---|---|---|---|
| CSC | 0.90 | 0.54 | 0.937 |
| CC | 0.78 | 0.47 | 0.858 |
| PC | 0.77 | 0.45 | 0.850 |
| AQ | 0.79 | 0.49 | 0.891 |
| BL | 0.88 | 0.72 | 0.881 |
| BAS | 0.90 | 0.65 | 0.902 |
| PQ | 0.79 | 0.65 | 0.787 |

For reliability analysis, Cronbach's Alpha results show that the reliability values of both the INQ and BE scales are above 0.70. Thus, the CFA results and Cronbach's Alpha results show that the scales are consistent with research, accurate, and reliable.

Then, the Pearson correlation analysis showed that we examined the significance of the relationships between the first-order sub-dimensions of INQ (independent variable) and the first-order sub-dimensions of BE (dependent variable). Thus, the results showed that the correlation coefficients between INQ and its sub-dimensions and BE and its dimensions were statistically significant ($p < 0.01$) (Table 7) [67].

**Table 7.** Pearson correlation analysis results.

| Variables | CSC | CC | PC | AQ | BL | BAS | PQ |
|-----------|-----|-----|-----|-----|-----|-----|-----|
| CSC | 1 | 0.901 ** | 0.910 ** | 0.916 ** | 0.786 ** | 0.851 ** | 0.783 ** |
| CC | | 1 | 0.873 ** | 0.880 ** | 0.747 ** | 0.817 ** | 0.744 ** |
| PC | | | 1 | 0.887 ** | 0.755 ** | 0.820 ** | 0.766 ** |
| AQ | | | | 1 | 0.746 ** | 0.817 ** | 0.760 ** |
| BL | | | | | 1 | 0.893 ** | 0.820 ** |
| BAS | | | | | | 1 | 0.843 ** |
| PQ | | | | | | | 1 |

** $p < 0.01$, CSC: Channel-Service Configuration, CC: Content Consistency, PC: Process Consistency, AQ: Assurance Quality, BL: Brand Loyalty, BAS: Brand Awareness and Brand Association, PQ: Perceived Quality.

We used the Fornell–Larcker Criterion (FLC) to calculate the discriminant validity of the scale with discriminant validity analysis [68]. The discriminant validity of our constructs was ensured because the square roots of the extracted mean variance (AVE) values of the scale variables of our study were higher than the correlations between the constructs [68]. Thus, the discriminant validity of the scale of the study was ensured (Table 8).

**Table 8.** Discriminant validity results.

| Variables | CSC | CC | PC | AQ | BL | BAS | PQ |
|-----------|-----|-----|-----|-----|-----|-----|-----|
| CSC | **0.93** | | | | | | |
| CC | 0.91 | **0.92** | | | | | |
| PC | 0.92 | 0.90 | **0.93** | | | | |
| AQ | 0.88 | 0.90 | 0.91 | **0.92** | | | |
| BL | 0.91 | 0.89 | 0.85 | 0.91 | **0.93** | | |
| BAS | 0.89 | 0.91 | 0.91 | 0.83 | 0.89 | **0.89** | |
| PQ | 0.89 | 0.90 | 0.90 | 0.90 | 0.89 | 0.88 | **0.87** |

Notes: The square root of AVE is indicated in bold; CSC: Channel-Service Configuration, CC: Content Consistency, PC: Process Consistency, AQ: Assurance Quality, BL: Brand Loyalty, BAS: Brand Awareness and Brand Association, PQ: Perceived Quality.

*5.3. Result for Structural Models*

For the analysis of our hypotheses, we used the method SEM (structural equation modeling), which is hypothesized due to its compatibility with the research, is recommended for analysis with small or medium sample sizes, and does not require a normal distribution of the data. For SEM, we used the program LISREL 8.7 [44,64,146,147].

SEM analysis values of hypothesis testing were found in two different models. The first path diagram includes each degree of relationship between INQ and the dimensions BL, PQ, and BAS separately (Figure 2). The statistical results showed a standardized beta of 0.86 (INQ-BL), 0.93 (INQ-PQ), and 0.93 (INQ-BAS), respectively (Figure 3). These values were significant at $p < 0.001$ (Table 9). According to [148], the $R^2$ value was accepted as a significant effect size. Furthermore, the goodness-of-fit values were found to be excellent according to the literature: $X^2/df = 4.492$, RMSEA = 0.077, CFI = 0.99, IFI= 0.99, RMR= 0.041, SRMR = 0.040, GFI = 0.93, AGFI = 0.91, NFI = 0.99, NNFI = 0.99. Thus, hypotheses $H_1$, $H_2$, and $H_3$ were confirmed, which stated that overall integration quality had a significant and positive influence in the sub-dimensions of BL, perception quality, and BAW/BA (BAS), which are combined into one dimension.

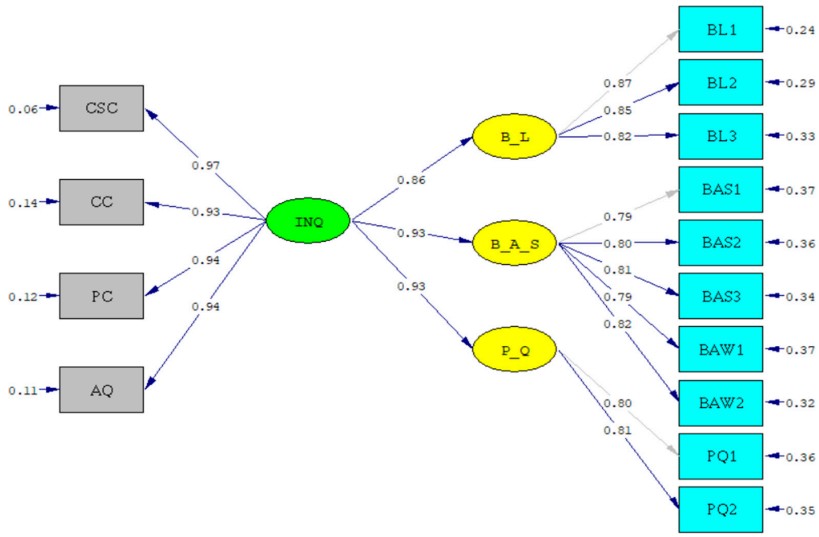

Chi-Square=342.53, df=73, P-value=0.00000, RMSEA=0.077

**Figure 2.** The first structural model.

**Table 9.** Findings for the first structural model.

| Paths | β/Path Coefficients | T Statistics | *p* Value | Result |
|---|---|---|---|---|
| INQ-BL | 0.86 | 22.39 ** | 0.000 | Accepted |
| INQ-PQ | 0.93 | 22.11 ** | 0.000 | Accepted |
| INQ-BAS | 0.93 | 21.66 ** | 0.000 | Accepted |

** *p* < 0.01.

The other path diagram shows the relationship between INQ and brand equity within the structural model. (Figure 3). The standardized beta value for INQ-BE was 0.90, and *p* < 0.001 was significant (Table 10). The model explained the total variance by $R^2$. According to [148], the $R^2$ value was accepted as a significant effect size. Thus, hypothesis $H_4$ was confirmed, which states that the overall quality of integration has a significant and positive effect on brand equity.

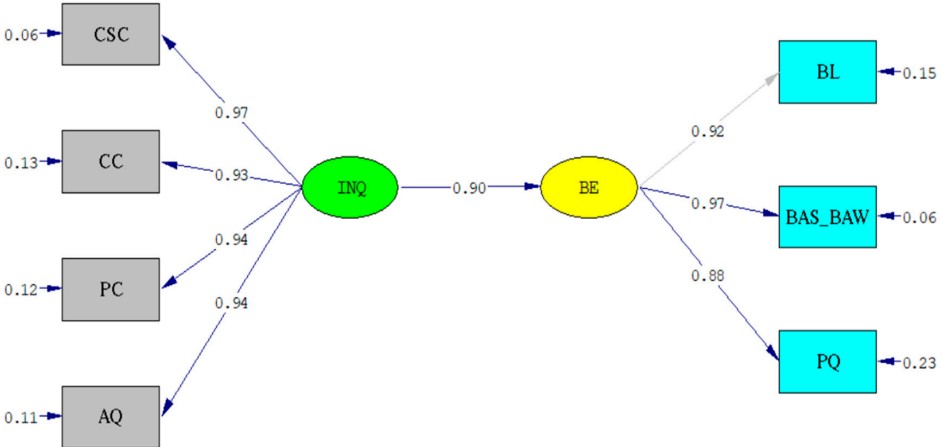

Chi-Square=28.51, df=13, P-value=0.00000, RMSEA=0.044

**Figure 3.** The second structural model.

**Table 10.** Findings for the second structural model.

|  | β/Path Coefficients | T Statistics | *p* Value | Result |
|---|---|---|---|---|
| INQ-BE | 0.90 | 25.56 ** | 0.000 | Accepted |

Source: From the authors. ** *p* < 0.01.

## 6. Discussion and Implications

Based on the dimensions of integration quality proposed by Hossain et al., in 2020, we presented a model of the brand effect of INQ and tested it empirically using SEM. While INQ consists of four sub-dimensions, namely channel-service configuration, content consistency, process consistency, and assurance quality, brand equity consists of four sub-dimensions: BL, PQ, BAW, and Brand Association. We confirmed with CFA that the size of the scales decreased to one, which is consistent with the literature.

The results of the structural models have confirmed that INQ is an important predictor and promoter of brand equity and its sub-dimensions (BL, PQ, BAW, and BA combined in one dimension). In other words, our research proves that brand equity and its dimensions (BL, PQ, BAW and BA-(BAS)) for Nike, a luxury sportswear brand that uses omnichannel, are a result of integration quality, which is a performance indicator of the harmony of all services provided at all customer touchpoints of the brand. For this reason, this research has revealed that brands that use omnichannel marketing care about integration quality and have higher brand equity than other brands competing in the same market that do not care about omnichannel marketing integration quality.

Considering the results of the first hypothesis test from the research model results in Table 10, it was found that integration quality had a positive and significant effect on BL (t = 22.39, *p* < 0.01). When the level of integration quality increases by one unit in the channels of Nike, a luxury sportswear brand that engages in omnichannel marketing, the loyalty level of the brand is positively affected by 0.86. This relationship between INQ and BL has not been previously examined in the omnichannel study. However, in a multichannel environment, the relationship between the factor of service quality used instead of INQ and BL has been established in a few studies [10,42,51,52]. For example, [149] research conducted in Malaysia found that service quality is effective in creating brand loyalty. However, unlike multichannel, this hypothesis finding makes a unique contribution to the literature, as omnichannel means delivering the same service to customers in all channels at the same time [10]. With the provision of cross-channel INQ in an omnichannel environment, customer satisfaction, which is an important antecedent of brand loyalty in the literature, can be gained. The common entity that is the factor in the emergence of OM is consumer desire. In this context, the service that will be provided with the same transparency, harmony and strategies in all channels, which will lead to an increase in customer satisfaction and thus brand loyalty [18,150].

When examining the result of the second hypothesis test, it was found that the integration quality of Nike brand's sales channels had a positive and significant effect on BA and BAW, which are combined into a single factor (t = 2.11, *p* < 0.01). Thereby, there will be a positive effect of 0.93 on BA and BAW when the integration quality increases by one unit, summarized in a single factor. Brand awareness and brand association components are actually influenced by brand image [16,17,106,108,151]. The image of a brand in society can change what consumers think about the brand. Therefore, if a brand has a good image, the consumer's awareness of the brand is positively affected. In this context, the positive effect of INQ, which means the performance measurement of the harmony to be achieved between channels, on brand association and awareness in this study offers an important opportunity for brands to improve their brand image, brand awareness and brand association for omnichannel environments [43,45,149]. In addition, it has been revealed in previous studies that communication has a great contribution on brand awareness. Based on this finding, inter-channel communication and one-to-one communication at contact points with customers will be beneficial both in increasing the level of INQ and in improving

brand awareness and attitude toward the brand [33,152,153]. In this context, this SEM result, which reveals the effect of INQ on brand awareness and brand association in the OM environment, opens the way for a fairly new research area and reduces the literature gap.

As a result of the third hypothesis test, it was found that the integration quality of the Nike brand distribution channels had a positive and significant effect on the quality of the brand perceived by the consumer (t = 21.66, *p* < 0.01). A one-unit increase in the level of integration quality attributable to the compliance performance of all brand channels has a positive effect of 0.93 on the quality that the brand evokes in the minds of consumers. The impact of service quality on the perceived quality of the brand was previously discussed for multi-channel environments [53,91,125,127,132,154]. However, to the best of the author's knowledge, we are not the first research to directly reveal this relationship, as there are already very limited studies for omnichannel. Some antecedents have been provided in the literature for a brand trying to increase its perceived quality in the minds of its customers, which is related to the excellence of a brand in the most general sense. This study obtained INQ as a predictor of the perceived quality of the brand. The finding of our research, which allows us to redefine an influencer of the concept of perceived quality, which is an abstractly felt quality, for the OM environments brought by digitalization, is a first in its field. For OM, there is no system that concretely measures the services offered through the channels. In other words, the approximation of the inter-channel compatibility performance to the maximum will lead to improvements in the perceived quality of the brand. Past research has presented evidence that perceived quality increases brand loyalty [22,130–132]. In addition to the satisfaction of customers with product quality, product, price, and promotions, purchasing the brand's products from all online and offline channels of the brand with the same quality, price, customer relations, after-sales service, discounts and payment facilities will increase the perceived quality of the brand.

Finally, we were curious to know whether brand equity, which is important as a component of customer equity, is affected by the integration quality of the brand's channels and the integration quality of a brand in the case of omnichannel marketing application. Considering the results of the first hypothesis test from the research model results in Table A1, in the fourth hypothesis, we examined the impact of integration quality on brand equity and found that integration quality had a positive and significant impact on brand equity (t = 25.57, *p* < 0.01). If the brand achieves an improvement of one unit in integration quality, which is an indicator of the harmonization performance of the operation in the brand's channels, the equality of this brand will increase by 0.90 units in the positive direction. In other words, we reduce the literature gap with this hypothesis, which was asked to be discussed by [45]. With this study, which explores the relationship between INQ and brand equity for the first time, we provide theoretical contributions and expand the explored research area in customer equity subject areas with INQ. Brand managers and marketers have historically resorted to many innovations to increase brand equity. Time intervals and innovations brought by digitalization have changed the strategies of 4P (Product, Price, Place, Promotion) while also offering new markets for brands to present their products. What did not change with all this was the effort to increase the value of the brand's tangible and intangible assets. We pave the way for future BE research with our research, where we present a very usable result for omnichannel environments to investigate the factors that may be effective in increasing BE.

### 6.1. Theoretical Implications

The findings of this study have some unique theoretical implications.

First, research on omnichannel applications is quite limited. This study brings a conceptual extension to the literature on omnichannel strategies, INQ, OM.

Second, this research has demonstrated that INQ is a provider of brand equity in an omnichannel environment that has thus far been tested with customer-oriented relationships and purchase intention, and as far as the authors know, its relationship with brand

assets has not been quantitatively studied. Therefore, it is the first research to examine these structures and relationships in the field of OM.

Third, the $H_1$, $H_2$, $H_3$ assumptions of this study, in which we set up an impact analysis with each of the INQ and BE dimensions (BL, BAS, PQ) were accepted. In this context, the work to discover the predecessors of BE has been extended with the INQ adopted for OM.

Fourth, we narrowed the literature gap in the context of BE by providing a validation for the antecedents of INQ (CSC, CC, PC, AQ), whose validation is not yet fully clarified in the literature.

### 6.2. Managerial Implications

This study has valuable contributions to marketers and brand managers. First of all, there are brands that are rapidly transitioning to omnichannel environments that have not yet been fully adopted, reaching customers through all channels, but experiencing losses compared to single-channel marketing in this direction. In this sense, we provided marketers with ways to increase INQ with this study. We also presented the necessary reasons for managers to provide INQ in an omnichannel environment in order to increase brand equity, with an empirical application. In addition, the most important benefit of this research is that it shows brand owners and managers and brand marketers a way to set up the omnichannel system.

### 7. Conclusions

In this research, we empirically investigated the relationships between INQ and BL, PQ, BAW, brand association and BE. The reliability, divergence and convergent validity results of the scale of our research exceeded the literature values. In other words, the scale we used for the empirical application of this research and the model of the research matched. Path coefficients and goodness-of-fit values obtained by SEM analysis recommended in the literature in impact studies were in agreement with the literature. Based on the problems of our research, four different hypotheses, which we constructed theoretically, were accepted. That is, the SEM results confirmed the assumptions we made between INQ and BE and their components in the context of OM. SEM results revealed that INQ had the highest effect (0.93) on BAS and PQ and the least effect (0.86) on BL, and INQ affected BE with 0.90.

The results of this research, which examine the predictors of brand equity and its components, have not been empirically analyzed despite increasing knowledge, and they offer implications for the subject areas OM, INQ, BE, which still have limitations in theoretical knowledge. Our research is unique, as it is the first study to empirically examine the relationship between INQ and BE and its components in the context of OM. It is also unique in that it is the first study to examine in detail the relationship between omni-channel strategies and brand equity in the context of Turkey, a developing country with very limited work on the Circular Revolution. Furthermore, our research results expand the limitations of some studies in the literature.

The results of INQ, which is one of the dynamics of service quality, and brand equity, which is a dimension of customer equity, will help managers, customers, companies, marketers and brand consultants to establish an effective omnichannel system. It will contribute to all available information. It has been revealed that the efforts of the brand to realize the integration of all services offered in the omnichannel environment have an effect on all information, awareness and associations in the mind of the consumer, the provision of INQ in this framework, the provision of the consumer's loyalty to the brand, the perceived quality of the brand in the mind of the customer, and the positive if it is negative, or already positive. If it is positive, it will have beneficial returns for the brand, such as reinforcement.

One of the broadest contributions of this article is that it offers a solution for omnichannel environments to reduce the cost of textile products returned due to inter-channel incompatibility, as well as the amount of waste generated by brands throwing away returned

products. Thus, we expand the field of research for the protection of the environment and reduce carbon emissions, as well as shed light on the research to be conducted in this field.

Our research has several limitations. First, the research data were collected only in Turkey. This limits the research, as the results cannot be generalized to other countries. In the research, the participants' data were collected in a single time frame in 2022, and the time limit is another limitation of the research. Furthermore, the research was only applied to Nike, a famous sportswear brand that operates in the apparel industry. Although generalization for the apparel sector is easy, the inability to extend the research to other sectors that require more services is the final limitation of the research. Expanding this research to overcome these three limitations is recommended for future studies. The impact of cross-channel integration quality on brand equity and dimensions can be studied for omnichannel in different income countries, periods, and industries. Moreover, the brand value was a dynamic of customer equity. Therefore, whether customer equity is an outcome of INQ could be a new research topic for the future.

**Author Contributions:** Conceptualization, T.Y. and M.I.; Formal analysis, T.Y.; Investigation, T.Y. and M.I.; Methodology T.Y. and M.I.; Project administration and supervision M.I.; Validation, T.Y. and M.I.; Writing—original draft, T.Y.; Writing—review and editing, T.Y. and M.I. All authors have read and agreed to the published version of the manuscript.

**Funding:** This research received no specific grant from any funding agency in the public, commercial, or not-for-profit sectors.

**Institutional Review Board Statement:** The study was conducted in accordance with the Social and Human Sciences Research Ethics Committee of Karabuk University, Turkey, and was approved by the Social and Human Sciences Research Ethics Committee of Karabuk University (2022/04 and 11 May 2022).

**Informed Consent Statement:** Not Applicable.

**Data Availability Statement:** Not applicable.

**Conflicts of Interest:** The authors declare no conflict of interest.

## Abbreviations

| | |
|---|---|
| OM | Omnichannel Marketing |
| INQ | Integration Quality |
| CSC | Channel-Service Configuration |
| CC | Content Consistency |
| PC | Process Consistency |
| AQ | Assurance Quality |
| BE | Brand Equity |
| BL | Brand Loyalty |
| BAW | Brand Awareness |
| BA | Brand Association |
| BAS | Brand Awareness and Association |
| PQ | Perceived Quality |
| CFA | Confirmatory Factor Analysis |
| SEM | Structural Equation Model |
| AVE | Subtracted Average Variance |
| CR | Composite Reliability |

# Appendix A

**Table A1.** Results of constructs.

| Constructs | Items | CR | AVE | Cronbach's Alpha |
|---|---|---|---|---|
| Channel-service choice [57,58] | This brand offers its products through multiple channels. | 0.90 | 0.54 | 0.93 |
| | When buying products of this brand, I can choose from more than one channel. | | | |
| | If I cannot buy products of this brand from a particular channel, I can use other channels. | | | |
| | I am aware of the products offered by this brand's multiple channels (website, physical store and mobile application). | | | |
| | I know how to use the features of this brand's multiple channels (website, physical store and mobile app). | | | |
| | This brand kept me well informed about the various features of its multiple channels (website, physical branch and mobile app). | | | |
| | This brand does not force me to use a particular channel for a particular purpose. | | | |
| | Services provided through different channels of this brand are suitable for these channels. | | | |
| Content Consistency [53,57,58] | The product prices of this brand are consistent across all channels. | 0.78 | 0.47 | 0.85 |
| | This brand provides consistent promotional information across different channels. | | | |
| | This brand provides consistent product information across different channels. | | | |
| | In general, the information on multiple channels of this brand is consistent. | | | |
| Process Consistency [57,58] | All channels of this brand (website, physical store and mobile application) are easy to use. | 0.77 | 0.45 | 0.85 |
| | All channels of this brand (website, physical store and mobile application) have a flexible system to meet my needs. | | | |
| | The service experience is consistent across all channels of this brand (website, physical store and mobile app). | | | |
| | This brand maintains a consistent brand image across all its channels (website, mobile app and physical store). | | | |
| Assurance quality [43] | My personal information is protected in all channels of this brand. | 0.79 | 0.49 | 0.89 |
| | My personal information in all channels of this brand isn't shared with others. | | | |
| | My financial information on all channels of this brand isn't shared with others. | | | |
| | All channels of this brand have sufficient security features. | | | |
| | All channels of this brand provide the means by which I can express my complaints. | | | |
| Brand Loyalty [22] | I plan to stay loyal to this brand. | 0.88 | 0.72 | 0.88 |
| | When purchasing a product, the products of this brand are always my first choice. | | | |
| | If there is a channel where I can reach this brand, I won't turn to other brands. | | | |
| Brand Association & Brand Awareness [22] | Some features of this brand come to my mind quickly. | 0.90 | 0.65 | 0.90 |
| | I can quickly remember the symbol or logo of this brand. | | | |
| | I have a hard time imagining this brand in my mind. | | | |
| | I easily recognize this brand among other competing brands. | | | |
| | I am aware of this brand. | | | |
| Perceived Quality [22] | The quality of the products of this brand is extremely high. | 0.79 | 0.65 | 0.78 |
| | The products of this brand are very likely to be functional. | | | |

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
