# Peer review of "Developing a Sustainable Omnichannel Strategic Framework toward Circular Revolution: An Integrated Approach"

_sustainability, doi:10.3390/su141811578_

Round 1
Reviewer 1 Report
Although this study has several advantages, it also has some areas that need correction.
1. In the section of introduction, the authors should provide key contributions (why the reader must read your article).
2. The data collection section is missing a lot of information. How did you reach out to so many potential respondents? What was the exact sampling criterion? When did you collect the data? Was there any response bias? What incentives were offered?
3. Discriminant validity results should be presented.
4. The citation style in the whole manuscript does not match the journal (Sustainability) at all.
5. A conclusion section should be added at the end of the paper.
Good luck.
Author Response
Dear reviewer, we greatly appreciate your review of this submission. With the aim of improving the manuscript, we have considered all your suggestions and made all suggested revisions. The revisions can be found in the revised manuscript as we used the track changes option and our response to each of your comments is attached.

Reviewer 2 Report
The paper examines an issue of considerable interest and significance. I have, however, a few comments and suggestions for them:
1. Introduction section will have to improve by giving the research gaps. I prefer a schematic diagram of the proposed approach so that audience can catch the novelty at a glance.
2. To show the actual novelty insert a Table at the end of the Introduction section showing related publications chronologically from the updated literature study. This part would basically focus on the hierarchy of literature study in brief showing your work’s novelty.
3. There is no conceptual comparison with existing approaches and no discussion on the benefits and drawbacks of the new approach. Thus discussions and comparative analyses should be added, also it is important to compare your method with the literature ones.
4. Conclusion section must be the last section. You can merge sections 9 and 10 together. Actually, there are too many sections in the present papers. The paper may be concluded within 5-6 sections. You could rearrange the sections accordingly.
5. Entries from Table 2 onwards are a little bit confusing. Please revise carefully.
6. There are too many redundant references in the Reference section that can be removed. Please add the following references and cite them in the text or future research directions: https://doi.org/10.1016/j.cie.2020.106765; https://doi.org/10.1007/s40819-020-0772-2
Author Response

(The authors gave the same response as above.)

Reviewer 3 Report
It is well written and discusses an interesting topic in the article "Developing Sustainable Omnichannel Strategic Framework for Circular Revolution: An Integrated Approach." A study of this nature is greatly needed. However, some issues need to be addressed.
A concise description of the purpose and implications of the study should be included in the abstract. Introductions should explain the study's context and research objective clearly. After analyzing the previous studies, the research gap must also be narrowed. The study's contribution to the field is not adequately claimed. The literature review needs to be improved. This study's results are relevant but have not been adequately discussed, nor have they been supported by significant and recent literature. It is difficult to justify their contribution when they used a limited and old dataset covering only 2004 to 2008. The authors should read recent and relevant publications. Finally, the authors should improve their conclusion and the quality of their work. The authors must focus on the conclusions supported by their findings rather than the recommendations they have outlined. The authors should discuss research directions and practical implications. There are some serious problems with English grammar and writing. It is recommended that authors hire a professional proofreader.
Author Response

(The authors gave the same response as above.)

Round 2
Reviewer 1 Report
The revision is now acceptable.
Author Response
Dear reviewer, we greatly appreciate your review of this submission. Thank youReviewer 2 Report
The authors have addressed the point of my concern. I am happy with their corrections. However, some fine-tuning is required. Some tables and Figures go outside the margin. For example Figure 1, Table 8, etc. Please take care.
Author Response

(The authors gave the same response as above.)
